KA-TP-04-2023
P3H-23-020
OUTP-23-03P

**LHCHWG-2022-004**

February 14, 2024

# LHC Higgs Working Group[a]

## Public Note

# Effective Field Theory descriptions of Higgs boson pair production

Lina Alasfar[1b], Luca Cadamuro[2c], Christina Dimitriadi[3,4d], Arnaud Ferrari[4e], Ramona Gröber[5f], Gudrun Heinrich[6g], Tom Ingebretsen Carlson[7h], Jannis Lang[6i], Serhat Ördek[4j], Laura Pereira Sánchez[7k], Ludovic Scyboz[8l] and Jörgen Sjölin[7m]

——————————————

[a] *https://twiki.cern.ch/twiki/bin/view/LHCPhysics/LHCHWG*

[b] alasfarl@physik.hu-berlin.de

[c] luca.cadamuro@cern.ch

[d] christina.dimitriadi@physics.uu.se

[e] arnaud.ferrari@physics.uu.se

[f] ramona.groeber@pd.infn.it

[g] gudrun.heinrich@kit.edu

[h] tom.ingebretsen-carlson@fysik.su.se

[i] jannis.lang@kit.edu

[j] serhat.oerdek@cern.ch

[k] laura.pereira.sanchez@cern.ch

[l] ludovic.scyboz@physics.ox.ac.uk

[m] sjolin@fysik.su.se

[1] *Institut für Physik, Humboldt-Universität zu Berlin, D-12489 Berlin, Germany*

[2] *Université Paris-Saclay, CNRS/IN2P3, IJCLab, 91405 Orsay, France*

[3] *Physikalisches Institut, Universität Bonn, 53115 Bonn, Germany*

[4] *Department of Physics and Astronomy, Uppsala University, 751 20 Uppsala, Sweden*

[5] *Dipartimento di Fisica e Astronomia "G.Galilei", Università di Padova and INFN, Sezione di Padova, 35131 Padova, Italy*

[6] *Institute for Theoretical Physics, Karlsruhe Institute of Technology, 76131 Karlsruhe, Germany*

[7] *Department of Physics, Stockholm University, Oskar Klein Center, SE-106 91 Stockholm, Sweden*

[8] *Rudolf Peierls Centre for Theoretical Physics, Clarendon Laboratory, Parks Road, University of Oxford, Oxford OX1 3PU, UK*

**Abstract**

Higgs boson pair production is traditionally considered to be of particular interest for a measurement of the trilinear Higgs self-coupling. Yet it can offer insights into other couplings as well, since – in an effective field theory (EFT) parameterisation of potential new physics – both the production cross section and kinematical properties of the Higgs boson pair depend on various other Wilson coefficients of EFT operators. This note summarises the ongoing efforts related to the development of EFT tools for Higgs boson pair production in gluon fusion, and provides recommendations for the use of distinct EFT parameterisations in the Higgs boson pair production process. This document also outlines where further efforts are needed and provides a detailed analysis of theoretical uncertainties. Additionally, benchmark scenarios are updated. We also re-derive a parameterisation of the next-to-leading order (NLO) QCD corrections in terms of the EFT Wilson coefficients both for the total cross section and the distribution in the invariant mass of the Higgs boson pair, providing for the first time also the covariance matrix. A reweighting procedure making use of the newly derived coefficients is validated, which can be used to significantly speed up experimental analyses.

# Chapter 1

# Introduction

Since the discovery of the Higgs boson a decade ago [1–3], the couplings to gauge bosons and third generation fermions have been measured to $\mathcal{O}(10 - 20\%)$ precision [4–8]. While these couplings give a good indication that the Higgs boson indeed behaves as predicted in the Standard Model (SM), an ultimate test of the mechanism of electroweak (EW) symmetry breaking is the measurement of the Higgs boson self-coupling.

The trilinear Higgs self-coupling can be measured in Higgs boson pair production. The dominant process at the Large Hadron Collider (LHC) is gluon fusion, which at leading order (LO) is mediated by triangle and box diagrams with loops of heavy quarks. The cross section is $\sim 31$ fb at $\sqrt{s} = 13$ TeV [9–19] and, as such, about three orders of magnitude smaller than the one for single Higgs boson production. To date, the most stringent bounds on the modification of the trilinear Higgs boson self-coupling, $\kappa_\lambda = \lambda_{hhh}/\lambda_{hhh}^{\text{SM}}$,[1] are provided by the ATLAS collaboration based on the LHC Run 2 dataset and are $-0.4 < \kappa_\lambda < 6.3$ [20]. The most recent constraints set by the CMS collaboration are $-1.2 < \kappa_\lambda < 6.5$ [8].

Usually two categories of new signatures in experimental searches for beyond the SM (BSM) physics in Higgs boson pair production are considered. In the first scenario, one expects that a relatively light new degree of freedom is exchanged and decays resonantly into a Higgs boson pair. In the second class of signatures, non-resonant Higgs boson pair production occurs, where the (heavy) new physics is parameterised in terms of operators and Wilson coefficients in an EFT framework. This note considers the latter case, where effective operators can modify the dominant gluon fusion Higgs boson pair production process in several ways. For instance, they allow for deviations

---

[1]Note that $\kappa_\lambda = c_{hhh}$, see Table 3.1 for coupling notation conventions.

in the top Yukawa coupling, can lead to a new coupling of two top quarks to two Higgs bosons, allow for an effective coupling of Higgs bosons to gluons and modify the trilinear Higgs self-coupling.

We start by reviewing the different EFT formulations (Chapter 2). In Chapter 3, we recap the state-of-the-art predictions for the Higgs boson pair production in the Standard Model Effective Field Theory (SMEFT) and Higgs Effective Field Theory (HEFT), as well as the available theoretical tools. We discuss in detail the theoretical uncertainties associated to those predictions. In experimental analyses, EFT limits are often set based on kinematical benchmarks, as illustrated for example in Refs. [21, 22]. In Chapter 4 we update the current ones [23–25] to accomodate bounds from single Higgs, and in particular $t\bar{t}H$, search results. Furthermore, we discuss the possibility of obtaining bounds on the EFT parameter space by making use of a reweighting procedure, which aims to considerably reduce the number of needed simulated events for experimental analyses. Finally, we conclude in Chapter 5.

# Chapter 2

# Effective Field Theory for Higgs boson pair production

We distinguish between two different kinds of EFTs with different assumptions made on the Higgs field, namely the SMEFT [26–29] and the HEFT [30–35]. The latter is also referred to as the non-linear chiral EW Lagrangian in the literature. In SMEFT, the Higgs field is assumed to transform as an $\mathrm{SU}(2)_L$ doublet like in the SM. The effective Lagrangian then allows for all operators compatible with the symmetries of the SM. In the Higgs sector, the leading operators arise at the dimension-6 level. We define the SM Lagrangian as

$$
\begin{aligned}
\mathcal{L} = {} & (D_\mu \phi)^\dagger (D^\mu \phi) + \mu^2 |\phi|^2 - \lambda |\phi|^4 \\
& - \left( y_d \bar{q}_L \phi d_R + y_u \bar{q}_L \tilde{\phi} u_R + y_e \bar{\ell}_L \phi e_R + \mathrm{h.c} \right) - \frac{1}{4} B_{\mu\nu} B^{\mu\nu} \\
& - \frac{1}{4} W^a_{\mu\nu} W^{\mu\nu,a} - \frac{1}{4} G^a_{\mu\nu} G^{\mu\nu,a} + \sum_{\psi = q_L, \ell_L, u_R, d_R, e_R} i \bar{\psi} \slashed{D} \psi \, , \qquad (2.1)
\end{aligned}
$$

where $\tilde{\phi}_i = \epsilon_{ik} \phi_k^*$, $q_L$ and $\ell_L$ are the quark and lepton $\mathrm{SU}(2)_L$-doublets, $u_R$, $d_R$ and $e_R$ are the $\mathrm{SU}(2)_L$-singlets. A summation over the different generations of quarks and leptons is assumed implicitly. The $\mathrm{SU}(2)_L$ Higgs doublet field in the unitary gauge is given by $\phi = 1/\sqrt{2}(0, v + h)^\intercal$, with $v$ denoting the vacuum expectation value, $v \approx 246\,\mathrm{GeV}$. Finally, $G_{\mu\nu}$, $W_{\mu\nu}$ and $B_{\mu\nu}$ are the SU(3), SU(2) and U(1) field-strength tensors.

The effective Lagrangian at dimension-6 can generally be written in various bases, with the different operators connected by field redefinitions. Two different complete bases are the Warsaw basis [27] and the strongly-interacting light Higgs basis (SILH), originally proposed in Ref. [36] and

completed in Refs. [37–39]. In addition, in Ref. [40] the so-called HISZ subset of operators is presented. In the Warsaw basis, the effective operators relevant for Higgs boson pair production (neglecting the couplings to light fermions) are given by

$$
\begin{aligned}
\Delta\mathcal{L}_{\text{Warsaw}} = {} & \frac{C_{H,\square}}{\Lambda^2}(\phi^\dagger\phi)\square(\phi^\dagger\phi) + \frac{C_{HD}}{\Lambda^2}(\phi^\dagger D_\mu\phi)^*(\phi^\dagger D^\mu\phi) + \frac{C_H}{\Lambda^2}(\phi^\dagger\phi)^3 \\
& + \left(\frac{C_{uH}}{\Lambda^2}\phi^\dagger\phi\,\bar{q}_L\tilde{\phi}\,t_R + \text{h.c.}\right) + \frac{C_{HG}}{\Lambda^2}\phi^\dagger\phi\, G^a_{\mu\nu}G^{\mu\nu,a} \\
& + \frac{C_{uG}}{\Lambda^2}(\bar{q}_L\sigma^{\mu\nu}T^a G^a_{\mu\nu}\tilde{\phi}\,t_R + \text{h.c.})\,.
\end{aligned}
\tag{2.2}
$$

While the Warsaw basis is constructed such that derivative operators are systematically removed by equations of motion, two derivative Higgs interactions remain. They contain covariant derivatives rather than simple derivatives and hence cannot be removed by gauge-independent field redefinitions. In order to obtain a canonically normalised Higgs kinetic term, the standard field redefinition (in unitary gauge) is

$$
\phi = \frac{1}{\sqrt{2}}\begin{pmatrix} 0 \\ h(1 + v^2\frac{C_{H,\text{kin}}}{\Lambda^2}) + v \end{pmatrix}
\tag{2.3}
$$

with

$$
C_{H,\text{kin}} = \left(C_{H,\square} - \frac{1}{4}C_{HD}\right)\,.
\tag{2.4}
$$

This field redefinition, however, generates derivative Higgs self-interactions, $h(\partial_\mu h)^2$ and $h^2(\partial_\mu h)^2$. For easier comparison with other effective descriptions, one can instead use a gauge-dependent field redefinition (which transforms Goldstone/Higgs components in a different way). However, such a choice needs to be made with care. While the full gauge-dependent field redefinition is given for instance in Ref. [41], we only need the transformation of the Higgs boson field:

$$
h \rightarrow h + v^2\frac{C_{H,\text{kin}}}{\Lambda^2}\left(h + \frac{h^2}{v} + \frac{h^3}{3v^2}\right)\,.
\tag{2.5}
$$

This field redefinition hence leads to a dependence on $C_{H,\text{kin}}$ for all Higgs boson couplings.

The SILH Lagrangian instead can be written as

$$\Delta\mathcal{L}_{\text{SILH}} = \frac{\bar{c}_H}{2v^2}\partial_\mu(\phi^\dagger\phi)\partial^\mu(\phi^\dagger\phi) + \frac{\bar{c}_u}{v^2}y_t(\phi^\dagger\phi\,\bar{q}_L\tilde{\phi}t_R + \text{h.c.}) - \frac{\bar{c}_6}{2v^2}\frac{m_h^2}{v^2}(\phi^\dagger\phi)^3$$
$$+ \frac{\bar{c}_{ug}}{v^2}g_s(\bar{q}_L\sigma^{\mu\nu}G_{\mu\nu}\tilde{\phi}\,t_R + \text{h.c.}) + \frac{4\bar{c}_g}{v^2}g_s^2\phi^\dagger\phi\,G_{\mu\nu}^a G^{a\mu\nu}\,. \qquad (2.6)$$

A canonical definition of the Higgs kinetic term can be obtained by means of the field redefinition

$$h \rightarrow h - \frac{\bar{c}_H}{2}\left(h + \frac{h^2}{v} + \frac{h^3}{3v^2}\right), \qquad (2.7)$$

again leading to a dependence on $\bar{c}_H$ for all Higgs boson couplings. While the operators relevant for Higgs boson pair production are basically the same in the SILH and Warsaw bases, we have adopted different power counting rules of the coefficients in front of the operators. For Eq. (2.2) a purely dimensional power counting is used, while Eq. (2.6) reflects a UV assumption regarding the scaling of the operators, e.g. new physics generating an operator $\phi^\dagger\phi\,G_{\mu\nu}^a G^{a\mu\nu}$, usually stems from coloured new particles that couple with the strong coupling constant $\alpha_s$ to the gluons. In Ref. [36, 42] for instance the coefficient in front of this operator contains an extra $1/16\pi^2$ to reflect the loop-suppression of weakly coupled new physics to the effective Higgs gluon coupling. We note that in Eqs. (2.2) and (2.6) we have considered only CP-even operators[1] due to strong bounds on CP-violating operators and we have considered only modifications of the top quark Yukawa couplings. We note though that modifications of light quark Yukawa couplings can be probed in Higgs boson pair production, see Refs. [44–46].

Considering now the HEFT Lagrangian, the relevant terms for Higgs boson pair production are given by

$$\Delta\mathcal{L}_{\text{HEFT}} = -m_t\left(c_t\frac{h}{v} + c_{tt}\frac{h^2}{v^2}\right)\bar{t}t - c_{hhh}\frac{m_h^2}{2v}h^3 \qquad (2.8)$$
$$+ \frac{\alpha_s}{8\pi}\left(c_{ggh}\frac{h}{v} + c_{gghh}\frac{h^2}{v^2}\right)G_{\mu\nu}^a G^{a,\mu\nu}\,.$$

In contrast to Eqs. (2.2) and (2.6), the couplings of one and two Higgs bosons to fermions or gluons become decorrelated. We also note that the top quark chromomagnetic dipole operator is omitted (i.e. an operator like the one with Wilson coefficient $\bar{c}_{ug}$ in the SILH basis or $C_{uG}$ in the Warsaw basis).

---

[1]See Ref. [43] for Higgs boson pair production allowing for CP-violation.

In a weakly interacting UV completion, such a coupling would enter at the loop level [47] and hence effectively be associated with an extra suppression factor of $1/16\pi^2$. In contrast to the $\phi^\dagger\phi\, G^a_{\mu\nu}G^{a\mu\nu}$ operator that carries such a suppression as well, the dipole operator enters Higgs boson pair production only via loop diagrams and is therefore suppressed compared to all the other operators assuming a weakly interacting UV model [48]. Comparing the coefficients of the different operators in the Lagrangians, one can derive relations between the Wilson coefficients in the Warsaw basis, SILH and HEFT.

Such a translation is given in Table 2.1. However, it has to be used with great care, as the different EFT descriptions rely on different assumptions and therefore are not necessarily translatable into each other. As a consequence, an anomalous coupling configuration which is perfectly valid in HEFT can lie outside the validity range of SMEFT upon such a naive translation. Examples are given in Chapter 3.

| HEFT | SILH | Warsaw |
|------|------|--------|
| $c_{hhh}$ | $1 - \frac{3}{2}\bar{c}_H + \bar{c}_6$ | $1 - 2\frac{v^2}{\Lambda^2}\frac{v^2}{m_h^2}\, C_H + 3\frac{v^2}{\Lambda^2}\, C_{H,\mathrm{kin}}$ |
| $c_t$ | $1 - \frac{\bar{c}_H}{2} - \bar{c}_u$ | $1 + \frac{v^2}{\Lambda^2}\, C_{H,\mathrm{kin}} - \frac{v^2}{\Lambda^2}\frac{v}{\sqrt{2}m_t}\, C_{uH}$ |
| $c_{tt}$ | $-\frac{\bar{c}_H + 3\bar{c}_u}{4}$ | $-\frac{v^2}{\Lambda^2}\frac{3v}{2\sqrt{2}m_t}\, C_{uH} + \frac{v^2}{\Lambda^2}\, C_{H,\mathrm{kin}}$ |
| $c_{ggh}$ | $128\pi^2\bar{c}_g$ | $\frac{v^2}{\Lambda^2}\frac{8\pi}{\alpha_s}\, C_{HG}$ |
| $c_{gghh}$ | $64\pi^2\bar{c}_g$ | $\frac{v^2}{\Lambda^2}\frac{4\pi}{\alpha_s}\, C_{HG}$ |

Table 2.1: Leading order translation between different operator basis choices.

As HEFT is more general than SMEFT, couplings of two Higgs bosons to fermions or gluons can be varied in an uncorrelated way with respect to the corresponding couplings with a single Higgs boson. While being more general, this obviously also has the disadvantage that more barely constrained couplings enter into Higgs boson pair production, leading potentially to degeneracies in their determination. In Table 2.1 we also see that the translation between the Warsaw basis and the SILH basis or HEFT contains $\alpha_s$. Since $\alpha_s$ is a running parameter and, for Higgs boson pair production, is typically evaluated at a central scale $\mu_0 = m_{hh}/2$, a translation between Warsaw and SILH/HEFT couplings needs to consider this caveat.

This can be rectified by including the running of $C_{HG}$ at the order at which the running of $\alpha_s$ is considered, or by redefining

$$C_{HG} \to C'_{HG} = \frac{1}{\alpha_s(\mu)} C_{HG}. \qquad (2.9)$$

Finally, we would like to comment on the models which are realised by the different choices of EFT. Typically, HEFT is the correct choice in strongly-interacting models where the Higgs boson arises as a pseudo-Goldstone boson. Since HEFT does not assume that the Higgs boson transforms within a SM doublet, Goldstone boson scattering is not unitarised by the Higgs boson, which in turn implies that the HEFT description cannot stay valid for new physics at scales of $\Lambda > 4\pi v$. Generally speaking, HEFT assumes larger deviations from the SM. Many UV models that are generically described by HEFT tend to linearise in the limit at which the coupling deviations are small with respect to the SM. For instance, models like Minimal Composite Higgs Models, given the current coupling constraints, can be reasonably well described by a linear EFT (SMEFT). Another prime example for HEFT, the dilaton, in its simplest description, typically predicts too large coupling deviations in the gluon Higgs couplings [49] and hence also its description via HEFT is challenged. A further example for a UV realisation of HEFT is the singlet model in the strong coupling regime keeping the vacuum expectation value of the singlet close to the EW scale [50]. Yet, in the regime where HEFT should be the preferred description, the mixing between singlet and doublet Higgs fields is rather large, and hence again strongly constrained by single Higgs boson coupling measurements. In the limit where both the new mass scale, singlet mass and singlet vacuum expectation value decouple, the model is well described within SMEFT. In Ref. [51] the conditions that apply when the HEFT description needs to be used are discussed and models that require a HEFT description are presented. These models have in common that 50% or more of the mass of the new state that is supposed to be integrated out is acquired via the EW vacuum expectation value. A study of these models in the context of Higgs boson pair production still remains an open question. Nevertheless, one should keep in mind that HEFT for Higgs boson pair production is more general and that Higgs boson pair production is the place for probing potential decorrelation among couplings of one or two Higgs bosons to fermions or gauge bosons (see Ref. [52] for multi-Higgs-boson production from longitudinal vector bosons).

# Chapter 3

# HEFT and SMEFT tools

Higgs boson pair production in gluon fusion at NLO QCD with full top quark mass dependence has been calculated in Refs. [10, 11, 53, 54]. Recently, the NLO QCD corrections based on analytic expressions in a combined $p_T$– and high-energy expansion also have become available [55], allowing for fast variations of the top quark mass renormalisation scheme. The corrections calculated in [10, 53] have been implemented in the publicly available codes ggHH [56–58] and ggHH_SMEFT [59, 60]. The code ggHH is based on the non-linear EFT framework (HEFT) described in Chapter 2 and allows for the variation of all five Wilson coefficients relevant to this process up to chiral dimension four and NLO QCD. The ggHH_SMEFT code is based on SMEFT and will be described in more detail below. The application of the HEFT framework to Higgs boson pair production at NLO QCD has been worked out in Ref. [24], where NLO results were presented for the twelve LO benchmark points defined in Ref. [23]. In Ref. [25], shapes of the Higgs boson pair invariant mass distribution $m_{hh}$ were analysed in the 5-dimensional space of anomalous couplings using machine learning techniques to classify $m_{hh}$ shapes, starting from NLO predictions. This analysis resulted in seven NLO benchmark points. Some of these benchmark points have been updated in Ref. [59] to be compatible with current experimental constraints, while we will update here additional ones.

In the following we will mostly focus on the description of the ggHH [58] and ggHH_SMEFT [59] codes, as they are the only publicly available codes that include the full top quark mass dependence at NLO. Further publicly available codes are the MG5_AMC@NLO framework [61], where the SMEFT@NLO code [62] contains a large number of operators, including the chromomagnetic dipole operator [63, 64], however the process $gg \rightarrow HH$ is

only available at LO in SMEFT. The `fortran` code `HPAIR` [65] is based on the analytic LO calculation of the Higgs boson pair production process [65–68] and includes the NLO corrections in the heavy top quark limit (HTL) [9]. More recent implementations based on this code are capable of computing the NLO HTL cross section with dimension-6 operators in SMEFT and non-linear EFT [69]. Furthermore, the packages `SMEFTsim` [70, 71] and `SmeftFR` [72, 73], built on `FeynRules` [74], contain a complete set of dimension-6 operators in the Warsaw basis, but are limited to LO, containing couplings of Higgs bosons to gluons only in the $m_t \to \infty$ limit.

## 3.1   HEFT combined with NLO QCD corrections

Parts of this section have been adapted from Ref. [58]

The effective Lagrangian relevant to $gg \to HH$ in HEFT is given by Eq. (2.8), where the conventions are such that in the SM $c_t = c_{hhh} = 1$ and $c_{tt} = c_{ggh} = c_{gghh} = 0$. The diagrams which contribute at LO in an expansion in $\alpha_s$ and up to chiral dimension $d_\chi = 4$ are shown in Fig. 3.1.[1] They are composed of loop diagrams built from terms appearing already at LO ($d_\chi = 2, L = 0$) in the chiral counting (first row) and of tree-level diagrams built from the next order ($d_\chi = 4, L = 1$) in the chiral counting (second row), based on the expansion of the EW chiral Lagrangian in loop orders $L$, where $d_\chi = 2L + 2$,

$$\mathcal{L}_{d_\chi} = \mathcal{L}_{(d_\chi=2)} + \sum_{L=1}^{\infty} \sum_i \left( \frac{1}{16\pi^2} \right)^L c_i^{(L)} O_i^{(L)} \,. \tag{3.1}$$

The two-loop diagrams entering the virtual corrections in HEFT have been calculated with the same method as described in Refs. [10, 53].

Within the HEFT approach, different normalisation conventions for the anomalous couplings are considered in the literature. In Table 3.1 we summarise some conventions commonly used. We note that the `ggHH` code [58] described in the following uses the convention of Eq. (2.8).

The `ggHH` code can be downloaded from the web page

`http://powhegbox.mib.infn.it`

under `User-Processes-V2` in the `ggHH` process directory. An example input card (`powheg.input-save`) and a run script (`run.sh`) are provided in the `testrun` folder accompanying the code.

---

[1] For details about the chiral dimension counting we refer to Refs. [24, 75–78].

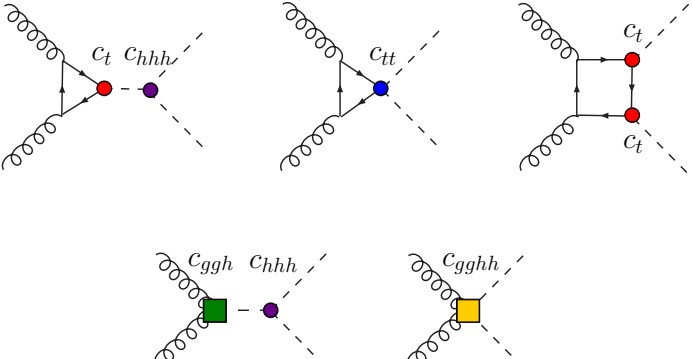

Figure 3.1: Higgs boson pair production in gluon fusion at LO in the chiral Lagrangian. The circles indicate vertices from anomalous couplings present already at leading chiral dimension ($d_\chi = 2$) in the Lagrangian, the squares denote effective interactions from contracted loops. Figure adapted from Ref. [79].

If all five anomalous couplings are varied, there is only the possibility of either calculating at NLO with full top quark mass dependence (`mtdep=3`), while at LO (setting `bornonly=1`) the five anomalous couplings can be varied either in the full theory or in the $m_t \to \infty$ limit. The approximations "Born-improved HTL" or "FT$_\text{approx}$" are available as additional options if only $c_{hhh}$ is varied.

The bottom quark is considered massless in all `mtdep` modes. The Higgs bosons are generated on-shell with zero width. Decays of the Higgs bosons can be considered through a parton shower (interfaces to Pythia 8 [80] and Herwig 7 [81] are contained in the code) in the narrow-width approximation. However, the decay is by default switched off (see the `hdecaymode` flag in the example `powheg.input-save` input card in `testrun`).

The masses of the Higgs boson and the top quark are set by default to $m_h = 125$ GeV and $m_t = 173$ GeV, respectively, and the top quark width is set to zero. The full SM two-loop virtual contribution has been computed with these mass values hardcoded, therefore they should not be changed when running with `mtdep = 3`, otherwise the two-loop virtual part would contain a different top quark or Higgs boson mass from the rest of the calculation. It is possible to change the values of $m_h$ and $m_t$ via the `powheg.input-save` input card when running with `mtdep` set to 0, 1 or 2.

The Higgs boson couplings can be varied directly in the `powheg.input`

| Eq. (2.8) | Ref. [23] | Ref. [69] |
|---|---|---|
| $c_{hhh}$ | $\kappa_\lambda$ | $c_3$ |
| $c_t$ | $\kappa_t$ | $c_t$ |
| $c_{tt}$ | $c_2$ | $c_{tt}/2$ |
| $c_{ggh}$ | $\frac{2}{3}c_g$ | $8c_g$ |
| $c_{gghh}$ | $-\frac{1}{3}c_{2g}$ | $4c_{gg}$ |

Table 3.1: Translation between different conventions for the definition of the anomalous couplings.

card. These are defined as follows, with their SM values as default:

`chhh=1.0:` the ratio of the Higgs trilinear coupling to its SM value,

`ct=1.0:` the ratio of the top quark Yukawa coupling to its SM value,

`ctt=0.0:` the effective coupling of two Higgs bosons to a top quark pair,

`cggh=0.0:` the effective coupling of two gluons to the Higgs boson,

`cgghh=0.0:` the effective coupling of two gluons to two Higgs bosons.

These are defined according to the Lagrangian of Eq. (2.8). The runtimes are dominated by the evaluation of the real radiation part. When run in the full NLO mode, the runtimes we observed for `POWHEG` stages 1 and 2 (i.e. the setup of the importance sampling grids and the estimation of the upper bounding envelope for `POWHEG`'s $\tilde{B}$ function) are in the ballpark of $100\,\mathrm{CPU}$ hrs for an uncertainty of about $0.1\%$ on the total cross section.

## 3.2 SMEFT and higher orders

Sections 3.2.1 and 3.2.2 have been adapted from Refs. [59, 82].

### 3.2.1 Discussion of truncation effects

The translation between HEFT and SMEFT is non-trivial. While the SILH Lagrangian and the Lagrangian in the Warsaw basis are conceptually very close, both describing an EFT where the Higgs sector is linearly realised, the HEFT approach relies on a different power counting scheme and therefore

higher orders in the EFT expansion are treated differently than in SMEFT, which relies on counting the canonical dimension in powers of $1/\Lambda$.

Comparing Eqs. (2.6) and (2.8), we derive the relations given in Table 2.1 (for $\Lambda = 1\,\mathrm{TeV}$). However, these relations hold at the level of the Lagrangian (expanded to a certain order in the EFT). Which terms to retain at amplitude squared, i.e. cross section level, is a subtle question.

The program `ggHH_SMEFT` [59] is a Monte Carlo (MC) program containing the full NLO QCD corrections to the process $gg \rightarrow HH$ as well as the effective operators relevant to this process within the SMEFT framework, up to canonical dimension-6 at Lagrangian level. The operator insertions are implemented in a modular way, allowing to study the truncation effects systematically. In order to construct the different truncation options we first decompose the amplitude into three parts: the pure SM contribution (SM), single dimension-6 operator insertions (dim6) and double dimension-6 operator insertions (dim6$^2$),

$$\mathcal{M} = \mathcal{M}_{\mathrm{SM}} + \mathcal{M}_{\mathrm{dim6}} + \mathcal{M}_{\mathrm{dim6}^2} \ . \tag{3.2}$$

For the squared amplitude forming the cross section, we consider four possibilities to choose which parts of $|\mathcal{M}|^2$ from Eq. (3.2) may enter:

$$\sigma \simeq \begin{cases} (a)\ \sigma_{\mathrm{SM}\times\mathrm{SM}} + \sigma_{\mathrm{SM}\times\mathrm{dim6}} \\ (b)\ \sigma_{(\mathrm{SM}+\mathrm{dim6})\times(\mathrm{SM}+\mathrm{dim6})} \\ (c)\ \sigma_{(\mathrm{SM}+\mathrm{dim6})\times(\mathrm{SM}+\mathrm{dim6})} + \sigma_{\mathrm{SM}\times\mathrm{dim6}^2} \\ (d)\ \sigma_{\left(\mathrm{SM}+\mathrm{dim6}+\mathrm{dim6}^2\right)\times\left(\mathrm{SM}+\mathrm{dim6}+\mathrm{dim6}^2\right)} \end{cases} \tag{3.3}$$

Option (a) is the first order of an expansion of the cross section $\sigma \sim |\mathcal{M}|^2$ in $\Lambda^{-2}$, sometimes also called linearised SMEFT. Option (b) is the first order of an expansion of the amplitude $\mathcal{M}$ in $\Lambda^{-2}$, which is then squared. The third option (c) includes all terms of $\mathcal{O}\left(\Lambda^{-4}\right)$ coming from single and double dimension-6 operator insertions, however it lacks the contributions at the same order from dimension-8 operators as well as $\mathcal{O}\left(\Lambda^{-4}\right)$ terms following the field redefinition of Eq. (2.3). Option (d) is the naive translation from HEFT to SMEFT using Table 2.1.

Typically, only the first two options are used for predictions based on SMEFT, the other options contain only a subset of operators contributing at dimension-8 and therefore are ambiguous. The recommendations concerning the application of the different options to experimental analyses is under discussion [83], and one of the purposes of the present note is to elucidate a few points related to this discussion. We included all of the options (a)-(d)

in our calculation, such that the effects of the different truncation options on the theory predictions can be studied in detail.

In the following we consider differential results, showing the effects of the different truncation options on the Higgs boson pair invariant mass distribution $m_{hh}$. We present results at benchmark points 1 and 6, given in Table 3.2, which are close to those presented in Ref. [25], based on an analysis of characteristic shapes of the $m_{hh}$ distribution.

| benchmark | $c_{hhh}$ | $c_t$ | $c_{tt}$ | $c_{ggh}$ | $c_{gghh}$ | $C_{H,\text{kin}}$ | $C_H$ | $C_{uH}$ | $C_{HG}$ |
|---|---|---|---|---|---|---|---|---|---|
| SM | 1 | 1 | 0 | 0 | 0 | 0 | 0 | 0 | 0 |
| 1 | 5.11 | 1.10 | 0 | 0 | 0 | 4.95 | $-6.81$ | 3.28 | 0 |
| 6 | $-0.68$ | 0.90 | $-\frac{1}{6}$ | 0.50 | 0.25 | 0.56 | 3.80 | 2.20 | 0.04 |

Table 3.2: Benchmark points used for the Higgs boson pair invariant mass distributions. The benchmark points derived in Ref. [25] were updated to accommodate new experimental constraints [7, 8, 20] (see Table 4.1 for the full set of new benchmark points). The value of $C_{HG}$ is determined using $\alpha_s(m_Z) = 0.118$. A value of $\Lambda = 1\,\text{TeV}$ is assumed for the translation between HEFT and SMEFT coefficients.

Figures 3.2 and 3.3 each show results for one benchmark at $\Lambda = 1\,\text{TeV}$ (upper panels), $\Lambda = 2\,\text{TeV}$ (middle panels) and $\Lambda = 4\,\text{TeV}$ (lower panels), for the different truncation options. The orange curve corresponds to the case (b), where squared dimension-6 contributions are taken into account, while the blue curve corresponds to the linear dimension-6 case. The envelope of a 3-point scale variation is shown for comparison for the SM and for case (b), as one of the viable SMEFT truncation options. We refrain from showing the scale uncertainties for the other curves, as their size would be similar, thus obscuring the figure. The negative differential cross section values in the linear dimension-6 case indicate that points in the coupling parameter space which are valid in HEFT can lead, upon naive translation, to parameter points for which the SMEFT expansion is not valid.

For benchmark point 6, the pattern of destructive interference between different parts of the amplitude (e.g. box- and triangle-type diagrams) in HEFT is similar to that in the SM case. However, in SMEFT (taking the squared dimension-6 level as reference), this interference pattern is modified, leading to a smaller cross section than in HEFT. Clearly, increasing $\Lambda$ reduces the differences between the results, as this corresponds to smaller deformations of the SM parameter space. Thus, for $\Lambda = 4\,\text{TeV}$ the different

truncation options appear almost indistinguishable at the precision of the presented plots (e.g. the orange and red distributions are covered by the dark green curve in the bottom row of Fig. 3.2). However, the characteristic shape (see Fig. 3.3) is not preserved for any of the considered $\Lambda$ values: in HEFT, the characteristic feature of benchmark 6 is a shoulder left of the main peak of the $m_{hh}$ distribution. In SMEFT, this shoulder is absent (except for option (d), the naive translation, and $\Lambda = 1\,\text{TeV}$, which corresponds to HEFT apart from the scale dependence of $\alpha_s$ in the Warsaw basis, see Table 2.1). Furthermore, we observe that the contribution from the interference of double dimension-6 operator insertions with the SM appears to be subdominant for benchmark point 1, but not for benchmark point 6, as can be seen by comparing the truncation option (b) in orange to the option (c) in red, the latter including the double operator insertions interfered with the SM amplitude.

Looking at the explicit values of the SMEFT coupling parameters in Table 3.2, stemming from the naive translation at $\Lambda = 1\,\text{TeV}$ between HEFT and SMEFT, it becomes clear that the parameters are too large for the SMEFT expansion up to dimension-6 to be valid. Therefore, the large differences seen in the results cannot be regarded as a truncation uncertainty. However, for $\Lambda = 2\,\text{TeV}$ these values are divided by a factor of 4, and even in this case linear dimension-6 is substantially different from quadratic dimension-6. In short, as Higgs boson pair production is a process with delicate cancellations between different parts of the amplitude, small differences in the treatment of the Wilson coefficients can have large effects.

In Ref. [83] various proposals of estimating the truncation uncertainty of the EFT expansion are discussed. With regards to Higgs boson pair production, we think that this uncertainty can be best estimated comparing the results obtained employing options (a) and (b) of Eq. (3.3) against each other (i.e. including only linear terms in $1/\Lambda^2$ at cross section level versus including the first order in $1/\Lambda^2$ in the amplitude and then squaring it).

We note also that Higgs boson pair production – due to its subtle interference structure – can show the major new physics effects at low invariant masses of the Higgs boson pair, as can be inferred for instance from the first benchmark scenario shown in Fig. 3.2. This is different from the typical assumption that effects from heavy new physics show up in the tails of the $m_{hh}$ distribution. Ref. [83] also proposes a procedure of estimating truncation effects by a *clipping* procedure, i.e. comparison of different results by employing different energy cuts $E_{\text{cut}}$. This would not necessarily provide information about the validity of the EFT expansion for Higgs boson pair production.

### 3.2.2 Usage of the program

The usage of the program `ggHH_SMEFT` [59] is very similar to that of the `ggHH` [58] code. Both are provided within the `POWHEG-BOX-V2` [84] under `User-Processes-V2`. The input card (`powheg.input-save`) allows to specify the values for `Lambda` (in TeV), `CHbox`, `CHD`, `CH`, `CuH` and `CHG`, with:

| | |
|---|---|
| `CHbox:` | the Higgs kinetic term coefficient $C_{H,\Box}$, |
| `CHD:` | the Higgs kinetic term coefficient $C_{HD}$, |
| `CH:` | the Higgs trilinear coupling term $C_H$, |
| `CuH:` | the Yukawa coupling to up-type quarks term $C_{uH}$, |
| `CHG:` | the effective coupling of gluons to Higgs bosons $C_{HG}$. |

The truncation options can be selected via the flag `multiple-insertion`, where the options (a)–(d) in Eq. (3.3) correspond to the values 0–3 of this flag. Otherwise the usage of the code is as described in Chapter 3.1. The chromomagnetic operator and four-top operators are subleading from the point of view of weakly interacting UV dynamics. These operators have been included in the `ggHH_SMEFT` code recently [60]. Considering the chromomagnetic operator and the four-top operators in isolation is delicate since the independence on the scheme to continue $\gamma_5$ to $D$ space-time dimensions can only be shown for a combination of these operators [85].

### 3.2.3 Theoretical uncertainties

Apart from the aforementioned uncertainties due to the truncation of the EFT, which can be estimated case by case using the options implemented in `ggHH_SMEFT`, there are further uncertainties associated to the computation of the cross section.

- Scale uncertainty

  Scale uncertainties are estimated by the variation of renormalisation and factorisation scales, $\mu_R = \mu_F = c \cdot \mu_0$, around the central scale $\mu_0 = m_{hh}/2$, with $c \in \{\frac{1}{2}, 1, 2\}$. In Figs. 3.2 and 3.3, the scale uncertainties have been assessed by a 3–point variation for the SM curve and for the SMEFT truncation option (b). At NLO, they are of the $\mathcal{O}(15\%)$ for the SM, and become $15 - 20\%$ (15%) for benchmark points 1 and 6 of Table 3.2 at $\Lambda = 1$ TeV ($\Lambda = 2$ or $4$ TeV). For the SM and benchmark point 1, it has been checked that the 7–point envelope agrees with the 3–point one. While the scale uncertainty is fairly symmetric

around the central value in the SM case, this does not necessarily hold for an arbitrary point in the EFT space. Scale uncertainties in HEFT have been assessed e.g. in Refs. [24, 58] and were found to be of similar magnitude. Including approximate NNLO corrections (calculated partly in the HTL) leads to a decrease of the scale uncertainties by a factor of 2 to 3 [79], depending on the benchmark point considered. However, as no public MC event generator is currently available at approximate NNLO to provide the scale uncertainties at an arbitrary coupling parameter point, we recommend to use NLO scale uncertainties, thereby considering a conservative uncertainty estimate.

- PDF+$\alpha_s$ uncertainty

  The SM PDF+$\alpha_s$ uncertainty at $\sqrt{s} = 13$ TeV and $\sqrt{s} = 14$ TeV amounts to $\pm 3\%$ at NNLO. It has been estimated with the Born-improved approximation using PDF4LHCNNLO [86] and found not to vary significantly with $c_{hhh}$ [87].[2] While the PDF+$\alpha_s$ uncertainty can be computed at NLO with the tools available, the uncertainty shrinks when going to NNLO and we do not expect this uncertainty to depend much on the chosen benchmark point. Hence, we recommend to include the SM uncertainty at NNLO.

- Top quark mass renormalisation scheme uncertainty

  The currently largest uncertainty on the SM cross section for Higgs boson pair production stems from the top quark mass renormalisation scheme. It has been obtained by forming the envelope between the NLO cross section calculated with the on-shell top quark mass and the $\overline{\text{MS}}$ top quark masses evaluated at the scales $\mu = m_t, m_{hh}$ and $m_{hh}/4$, and amounts to $^{+4\%}_{-18\%}$ [16] at $\sqrt{s} = 13$ TeV. Thus, the top quark mass renormalisation scheme uncertainty turns out to give the largest contribution of the uncertainty budget of Higgs boson pair production in the SM. We note that instead this uncertainty is rather small for on-shell single Higgs boson production, but it has been demonstrated recently that this uncertainty is also substantial for $gg \to ZH$, see [88, 89].[3] In Ref. [90], this uncertainty has been assessed for off-shell single

---

[2]However, one can assume that they would become slightly larger for EFT points where a major part of the cross section comes from large $m_{hh}$ values, as this would imply that the PDFs are evaluated at larger $x$.

[3]Furthermore, in Ref. [89] it has been shown that the uncertainty on the total cross section depends on the choice of the binning, due to the behaviour at the top quark mass threshold.

Higgs boson production, based on NNLO results with full top quark mass dependence for on-shell Higgs production [91] and the soft-virtual corrections at NNLO [92]. For off-shell single Higgs boson production it has been shown that, while the differences between the on-shell and $\overline{\text{MS}}$ schemes are sizeable, the predictions are always compatible within scale uncertainties. Ref. [90] also contains an assessment of the dependence of the trilinear Higgs self-coupling on the top quark mass renormalisation scheme in the limit where this coupling is very large, such that the triangle contributions dominate. The results indicate that the scheme uncertainties would be reduced at NNLO, and that the on-shell predictions have a better perturbative convergence.

The top quark mass renormalisation scheme uncertainty and the scale uncertainty have been combined linearly in Ref. [16], leading to a total uncertainty of $^{+6\%}_{-23\%}$ on the NNLO FT$_{\text{approx}}$ [15] SM cross section for Higgs boson pair production. The uncertainty also depends strongly on $c_{hhh}$ (e.g. for $c_{hhh} = -10$ it takes the values $^{+10\%}_{-6\%}$ [16]). This demonstrates that the uncertainty should be evaluated for each EFT parameter point separately. While this has been explicitly shown when varying the $c_{hhh}$ coupling, it becomes also clear for the other Wilson coefficients: for instance, one can assume that it becomes much smaller if the parameter point is driven by large $c_{ggh}$ or $c_{gghh}$, as this would reduce the relative dependence on the top quark mass. Unfortunately, the top quark mass scheme uncertainty is currently not available at full NLO for individual EFT parameter points. Hence, we leave it to future work to recommend its treatment in the EFT.

Furthermore, while in the SM the contribution of $b$-quark loops at LO is at the per-mil level, the inclusion of enhanced couplings to $b$-quarks, if considered in certain scenarios, would introduce another source of uncertainty due to the scale dependence of the $b$-quark $\overline{\text{MS}}$ mass.

- Uncertainty due to EW corrections

Partial results for the NLO EW corrections to Higgs boson pair production, related to the Yukawa-type corrections, have been calculated in Refs. [93, 94]. The NLO EW corrections to Higgs boson pair production in the large-$m_t$ limit have been calculated in Ref. [95], the full corrections recently also became available [96], resulting in a decrease of the total cross section by -4%, however for distributions the corrections can be larger in certain kinematic regions. NLO EW corrections in combination with HEFT or SMEFT are not available yet.

- Accuracy of the numerical computation of the NLO QCD virtual corrections

The NLO results obtained via the `ggHH` and `ggHH_SMEFT` codes feature two-loop virtual amplitudes computed numerically and assembled into a grid. The grid is interpolated such that the virtual amplitude can be evaluated at any phase-space point, and can be interfaced to an external MC integration program [56]. The phase-space points entering the grid have been sampled in order to obtain a rather uniform statistical accuracy, which is below $\lesssim 2\%$ in the original binning, in the distribution of $m_{hh}$ (up to $m_{hh} \sim 1.4$ TeV), in the SM. Nevertheless, one needs to be aware that for benchmark points associated with $m_{hh}$ shapes that are vastly different from the SM, that statistical uncertainty can increase. In particular, because the SM cross section is small in the first few bins above the $2\,m_h$ threshold, and thus the virtual grid is populated with very few points contributing to those bins, one can expect the statistical accuracy to be underestimated, particularly in benchmark points that present an enhancement of the low-$m_{hh}$ region.

We showcase this point in Fig. 3.4, for HEFT. In the upper plot, the SM differential cross section is plotted at NLO for $\sqrt{s} = 13$ TeV, along with the finite virtual contribution ($\mathcal{V}_{\text{fin}}$ [84] in `Powheg`), in the top panel. The statistical uncertainty associated with the sparseness of the virtual grid, with respect to the total cross section, is displayed in the bottom panel as a function of $m_{hh}$. As shown in the bottom panel, the final statistical accuracy is of the order of $\lesssim 2\%$, except in the first bin, where it reaches 12%. This can be traced back to the fact that the virtual grid is extremely sparse in this region, with only one kinematic point contributing to the first bin. This is exacerbated when we perform the same comparison for benchmark point 1 (which has a very enhanced low-$m_{hh}$ cross section due to the value of $c_{hhh}$, see the lower plot of Fig. 3.4). We indeed observe that the statistical uncertainty increases to about 70% in the very first $m_{hh}$ bin. The size of this statistical effect depends on the cross section, on the relative contribution of virtual corrections to the cross section in those bins, and on the size of the binning itself. Thus it cannot be estimated straightforwardly, *a priori*.[4] In particular, the binning should be cho-

---

[4]Consequently, the MC uncertainty calculated by `Powheg` in the `ggHH` and `ggHH_SMEFT` codes should not be taken at face value, in those bins. We note that the problem with the two-loop amplitude for non-SM parameters that has been detected after comparison with the authors of Ref. [55] has been fixed in revisions 4037 and 4038 of the `ggHH_SMEFT` and

sen carefully so as to avoid such poor statistical precision. Across the large set of EFT points produced for Ref. [79], where the first bin is defined as $[250, 290]$ GeV, the largest statistical uncertainty is $\mathcal{O}(12\%)$. The same effect appears, to a smaller extent, at large values of $m_{hh}$, for benchmark points which feature an enhanced tail (typically associated with large values of $c_{ggh}$ and $c_{gghh}$). The largest statistical uncertainty is $\mathcal{O}(4\%)$.

This issue may be alleviated by combining the two-loop amplitudes from the virtual grid with a low-$p_T$ expansion, respectively a high-energy expansion, in their regions of validity [97–99].

---

ggHH (HEFT) codes. All figures presented here are based on the corrected code.

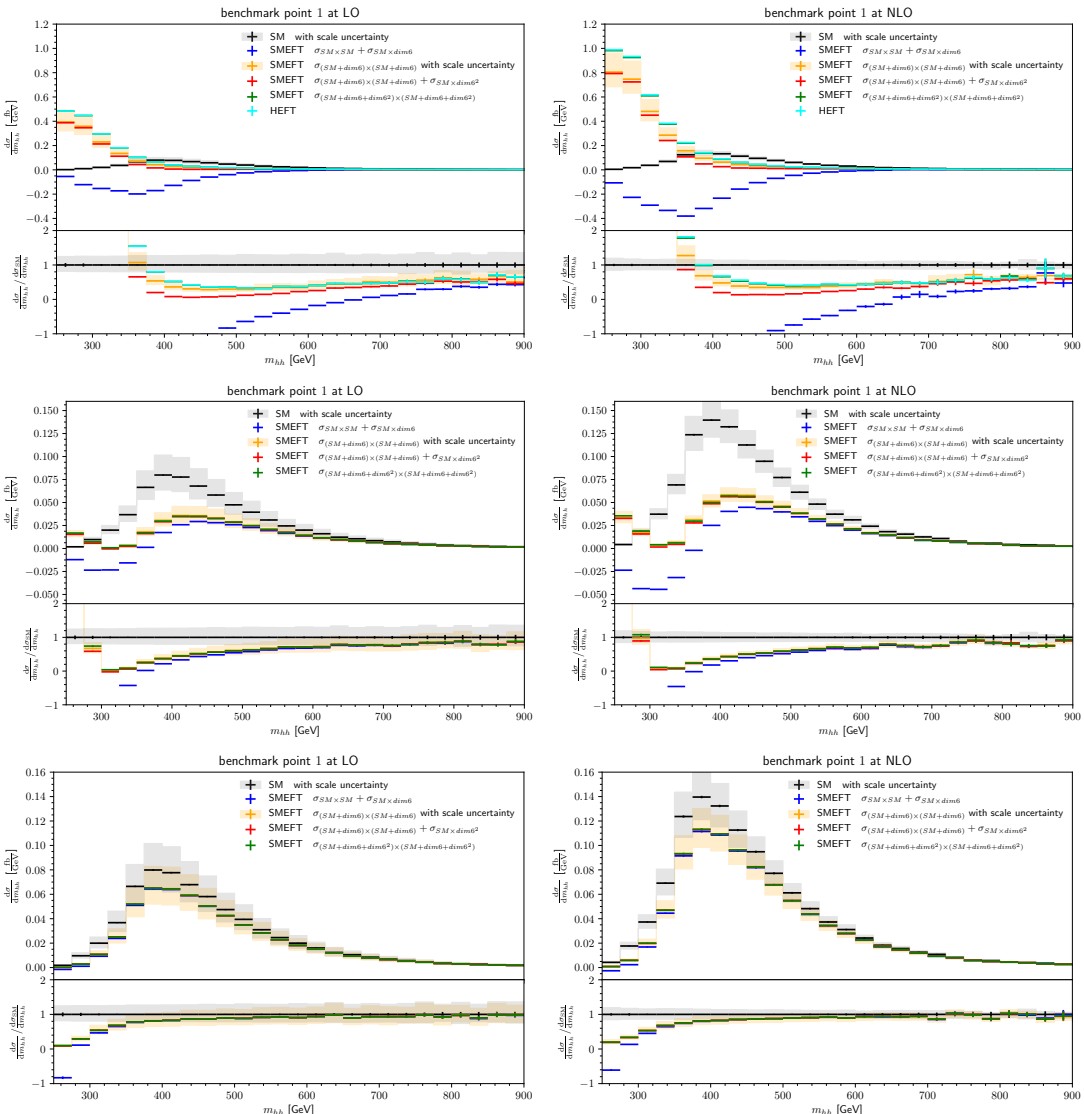

Figure 3.2: Differential cross sections for the invariant mass $m_{hh}$ of the Higgs boson pair for benchmark point 1 of Table 3.2. Top row: $\Lambda = 1\,\mathrm{TeV}$, middle row: $\Lambda = 2\,\mathrm{TeV}$, bottom row: $\Lambda = 4\,\mathrm{TeV}$. For the latter $\Lambda$ value the red and orange curves are almost indistinguishable form the dark green ones. The middle and bottom rows do not include a HEFT curve because the translation relies on $\Lambda = 1\,\mathrm{TeV}$. For benchmark point 1, option (d) in dark green and HEFT in cyan coincide because the only difference between these two comes from the running of $\alpha_s$ in the $C_{HG}$-term, however $C_{HG}$ is zero for this benchmark point. Left: LO, right: NLO. Figure adapted from Ref. [59].

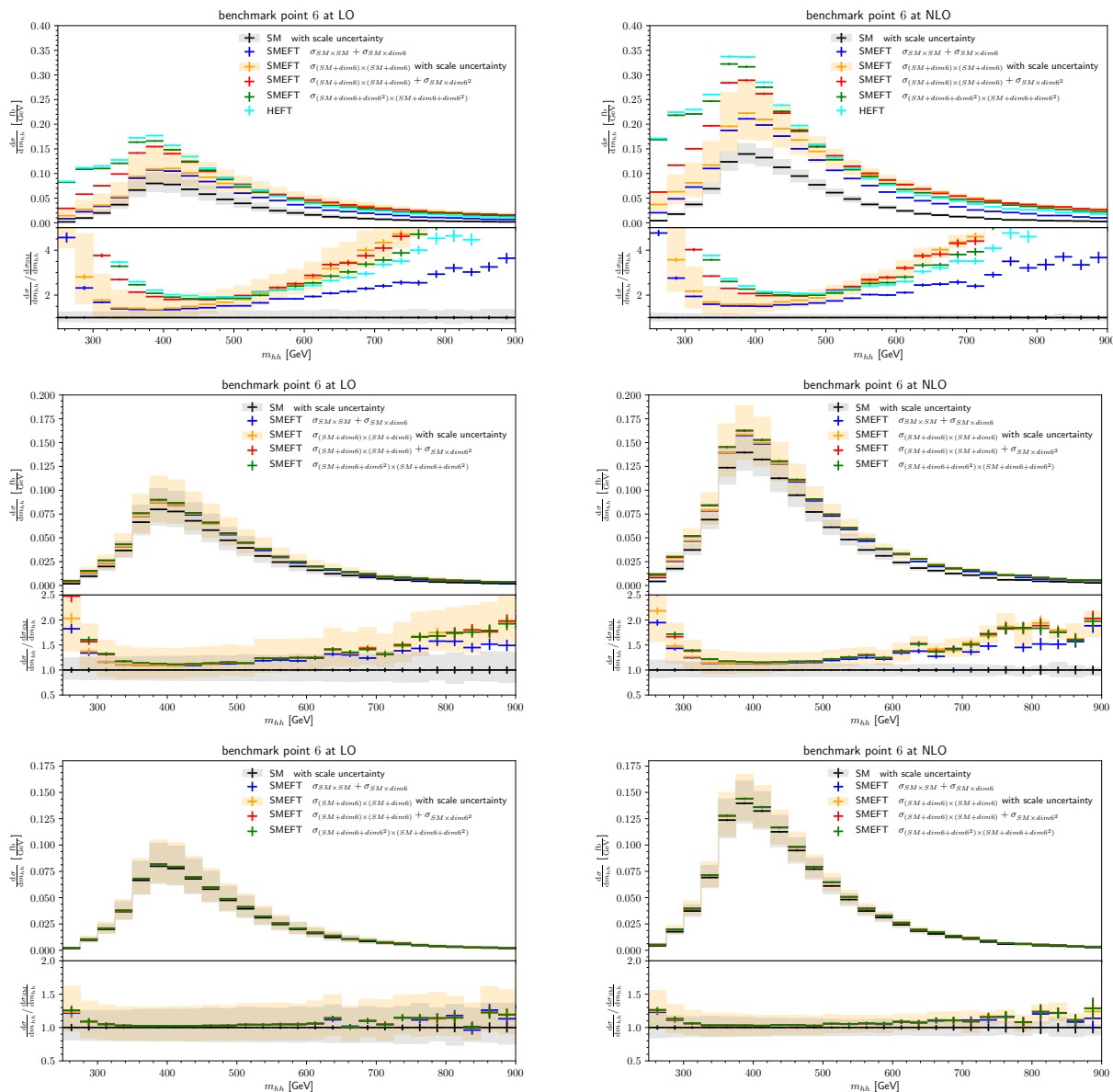

Figure 3.3: Differential cross sections for the invariant mass $m_{hh}$ of the Higgs boson pair for benchmark point 6 of Table 3.2. Top row: $\Lambda = 1\,\mathrm{TeV}$, middle row: $\Lambda = 2\,\mathrm{TeV}$, bottom row: $\Lambda = 4\,\mathrm{TeV}$. Left: LO, right: NLO. The middle and bottom rows do not include a HEFT curve because the translation relies on $\Lambda = 1\,\mathrm{TeV}$. Figure adapted from Ref. [59].

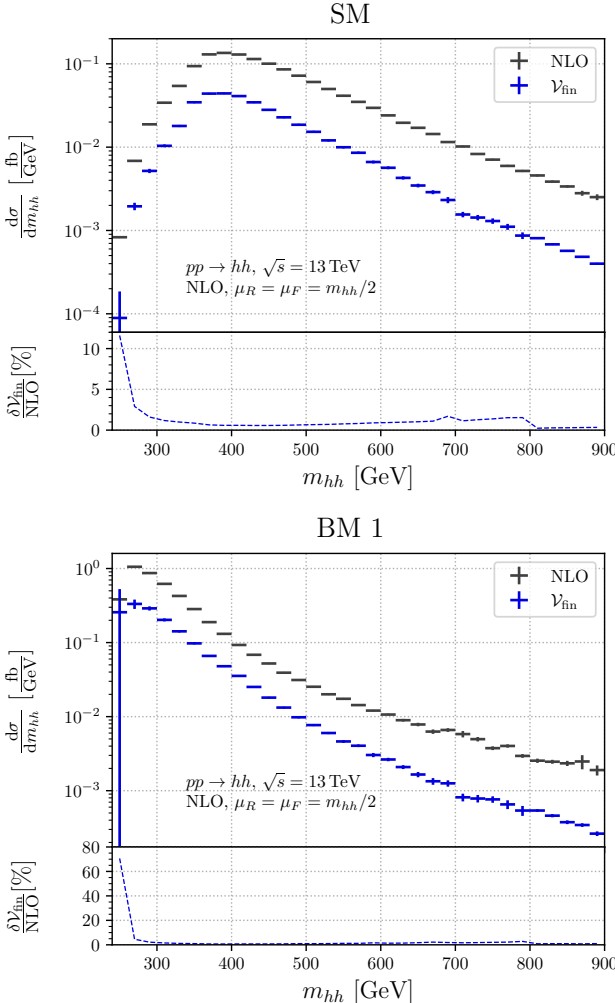

Figure 3.4: The differential cross section in $m_{hh}$ for the SM (top plot) and benchmark point 1 (bottom plot) along with the finite virtual contribution as calculated from the discrete virtual grid (blue) are shown. In the bottom panel, the statistical uncertainty on the total cross section, as propagated from the grid, is displayed as a function of $m_{hh}$.

# Chapter 4

# HEFT reweighting and validation

To avoid the computationally expensive simulation of multiple $HH$ samples, one can perform a reweighting of $HH$ events to any other point in the HEFT parameter space. The $HH$ production cross section ($\sigma_{hh}$) via gluon fusion can be parameterised for any set of HEFT Wilson coefficients at NLO as

$$
\begin{aligned}
\sigma_{hh}^{\mathrm{NLO}}(c_{hhh}, c_t, c_{tt}, c_{ggh}, c_{gghh}) = Poly(\mathbf{c}, \mathbf{A}) &= \mathbf{c}^{\mathsf{T}} \cdot \mathbf{A} \\
&= A_1 c_t^4 + A_2 c_{tt}^2 + (A_3 c_t^2 + A_4 c_{ggh}^2) c_{hhh}^2 \\
&+ A_5 c_{gghh}^2 + (A_6 c_{tt} + A_7 c_t c_{hhh}) c_t^2 \\
&+ (A_8 c_t c_{hhh} + A_9 c_{ggh} c_{hhh}) c_{tt} + A_{10} c_{tt} c_{gghh} \\
&+ (A_{11} c_{ggh} c_{hhh} + A_{12} c_{gghh}) c_t^2 \\
&+ (A_{13} c_{hhh} c_{ggh} + A_{14} c_{gghh}) c_t c_{hhh} \\
&+ A_{15} c_{ggh} c_{gghh} c_{hhh} + A_{16} c_t^3 c_{ggh} \\
&+ A_{17} c_t c_{tt} c_{ggh} + A_{18} c_t c_{ggh}^2 c_{hhh} \\
&+ A_{19} c_t c_{ggh} c_{gghh} + A_{20} c_t^2 c_{ggh}^2 \\
&+ A_{21} c_{tt} c_{ggh}^2 + A_{22} c_{ggh}^3 c_{hhh} \\
&+ A_{23} c_{ggh}^2 c_{gghh}
\end{aligned}
$$

(4.1)

where $\mathbf{A}$ is a set of coefficients determined from simulation and $\mathbf{c}^{\mathsf{T}}$ represents the vector of products of Wilson coefficients such that $\mathbf{c}^{\mathsf{T}} \cdot \mathbf{A} = Poly(\mathbf{c}, \mathbf{A})$. At LO only the first 15 terms of Eq. (4.1) are needed. In this publication we present a new set of coefficients $\mathbf{A}$ (for $\sqrt{s} = 13$ TeV), derived in a

similar way as in Ref. [24], but using a weighted least square fit. The new set of $\mathbf{A}$ coefficients predicts the cross section in pb, has a lower statistical uncertainty thanks to being derived using more simulated $HH$ MC events and covers a larger kinematic range.[1] In total 63 MC simulations are used including 62 HEFT samples (including the BM points in table 4.1) and one SM sample. The effect of BSM couplings on the kinematics of an $HH$ event can be approximated in terms of its effect on the Higgs boson pair invariant mass $m_{hh}$. Therefore, differential coefficients $d\mathbf{A}$ have also been derived for $m_{hh} \in [250, 1400]$ GeV with bins of 20 GeV for $m_{hh} \in [250, 1050]$ GeV and with two broader bins in the range $m_{hh} \in [1050, 1200]$ GeV and $m_{hh} \in [1200, 1400]$ GeV as

$$\frac{d\sigma_{hh}}{dm_{hh}}(c_{hhh}, c_t, c_{tt}, c_{ggh}, c_{gghh}) = Poly(\mathbf{c}, d\mathbf{A}|m_{hh}) = \mathbf{c}^\mathsf{T} \cdot d\mathbf{A}. \qquad (4.2)$$

The differential cross section can be used to reweight simulated $HH$ events (for example, SM with $\mathbf{c}_{\mathrm{SM}}$, i.e. $c_{hhh} = c_t = 1$ and $c_{tt} = c_{ggh} = c_{gghh} = 0$) to any other point of the HEFT parameter space as

$$w_{\mathrm{HEFT}} = \frac{Poly(\mathbf{c}, d\mathbf{A}|m_{hh})}{Poly(\mathbf{c}_{\mathrm{SM}}, d\mathbf{A}|m_{hh})}. \qquad (4.3)$$

These weights change both the shape of the $m_{hh}$ distribution and its normalisation by a factor $\sim \sigma_{hh}/\sigma_{hh}^{\mathrm{SM}}$. Given the limited range of $m_{hh} \leq 1400$ GeV of the coefficients $d\mathbf{A}$ and the larger statistical uncertainty in the derivation of individual $d\mathbf{A}$ coefficients, the total cross section is better predicted by the inclusive $\mathbf{A}$ coefficients. Therefore, it is more precise to use the inclusive values to determine the overall normalisation of $HH$ event distributions. When using the $d\mathbf{A}$ coefficients for MC reweighting, simulated $HH$ events with $m_{hh} > 1400$ GeV should be assigned the weight of the highest available $m_{hh}$ bin, i.e. of the [1200, 1400] GeV bin. This is not exactly correct, but it is the best available estimate. The highest existing precision of the parameterised $\sigma_{hh}$ is at approximate NNLO, the coefficients at $\sqrt{s} = 14\,\mathrm{TeV}$ have been derived in Ref. [79]. Here we show the reweighting based on $\mathbf{A}$ coefficients for $\sqrt{s} = 13\,\mathrm{TeV}$ at NLO.

We provide three sets of coefficients with scale variations of $\mu_R = \mu_F = c \cdot \mu_0$ with $\mu_0 = m_{hh}/2$ and $c \in \{\frac{1}{2}, 1, 2\}$, which can be used to derive continuous scale systematic uncertainties. The covariance matrices for the $\mathbf{A}$ and $d\mathbf{A}$ coefficients are also provided. These can be used to obtain the

---

[1]Ref. [24] provides differential coefficients up to 1040 GeV in $m_{hh}$.

statistical uncertainty on $Poly(\mathbf{c}, \mathbf{A})$ in Eq. (4.1) and $Poly(\mathbf{c}, d\mathbf{A}|m_{hh})$ in Eq. (4.2) via

$$\delta_{Poly(\mathbf{c},\mathbf{A})} = \sqrt{\mathbf{c}^{\mathsf{T}} \Sigma_{\mathbf{A}} \mathbf{c}} \tag{4.4}$$

and

$$\delta_{Poly(\mathbf{c},d\mathbf{A}|m_{hh})} = \sqrt{\mathbf{c}^{\mathsf{T}} \Sigma_{d\mathbf{A}} \mathbf{c}} \, , \tag{4.5}$$

where $\Sigma_{\mathbf{A}}$ and $\Sigma_{d\mathbf{A}}$ are the covariance matrices for $\mathbf{A}$ and $d\mathbf{A}$. With these, the statistical uncertainty for $w_{\mathrm{HEFT}}$ in Eq. (4.3) is calculated as

$$\delta_{w_{\mathrm{HEFT}}} = \sqrt{\mathbf{J}_w \Sigma_{d\mathbf{A}} \mathbf{J}_w^{\mathsf{T}}}, \tag{4.6}$$

where $\mathbf{J}_w$ is the Jacobian acting on Eq. (4.3) and has the following form

$$\mathbf{J}_w = \frac{\mathbf{c}^{\mathsf{T}}}{Poly(\mathbf{c}_{\mathrm{SM}}, d\mathbf{A}|m_{hh})} - \frac{Poly(\mathbf{c}, d\mathbf{A}|m_{hh}) \cdot \mathbf{c}_{\mathrm{SM}}^{\mathsf{T}}}{Poly(\mathbf{c}_{\mathrm{SM}}, d\mathbf{A}|m_{hh})^2}. \tag{4.7}$$

The total statistical uncertainty in bin $j$ when reweighting simulated SM $HH$ events is as follows:

$$\delta^j = N^j \sqrt{\left(\frac{\delta^j_{w_{\mathrm{HEFT}}}}{w^j_{\mathrm{HEFT}}}\right)^2 + \left(\frac{\delta^j_{\mathrm{SM}}}{N^j_{\mathrm{SM}}}\right)^2}, \tag{4.8}$$

where $N^j$ is the sum of weighted events, $N^j_{\mathrm{SM}}$ is the sum of weighted SM events, $w^j_{\mathrm{HEFT}}$ is the weight and $\delta^j_{\mathrm{SM}}$ is the weighted statistical uncertainty for the SM $HH$ events in bin $j$.

In the following, we show a validation of this reweighting procedure for a set of seven benchmark points which were originally identified in Ref. [25], based on a clustering of characteristic $m_{hh}$ shapes at NLO in HEFT with the help of unsupervised machine learning. Here, we update the benchmark points derived in Ref. [25] to take into account recent experimental constraints on some anomalous couplings [7,8]. More precisely, we have applied the same clustering of characteristic $m_{hh}$ shapes as in Ref. [25], with the tighter constraint $0.83 \leq c_t \leq 1.17$ for all benchmark points, and $|c_{tt}| < 0.05$ for benchmark 1. The benchmark points 1 and 6 were updated already in Ref. [59], while the updated scenarios 2 and 4 are shown here for the first time. Figures 4.1 and 4.2 show the $m_{hh}$ distributions for a sample of $HH$ events generated assuming Wilson coefficient values corresponding to specific benchmark points, compared to SM simulated events which have been reweighted following Eq. (4.3).

The reweighting is also tested for the average transverse momentum of both Higgs bosons, $p_T(h)$, as shown in Figs. 4.3 and 4.4. While the general

| benchmark | $c_{hhh}$ | $c_t$ | $c_{tt}$ | $c_{ggh}$ | $c_{gghh}$ |
|:---:|:---:|:---:|:---:|:---:|:---:|
| SM | 1 | 1 | 0 | 0 | 0 |
| 1 | 5.11 | 1.10 | 0 | 0 | 0 |
| 2 | 6.84 | 1.03 | $\frac{1}{6}$ | $-\frac{1}{3}$ | 0 |
| 3 | 2.21 | 1.05 | $-\frac{1}{3}$ | $\frac{1}{2}$ | $\frac{1}{2}$ |
| 4 | 2.79 | 0.90 | $-\frac{1}{6}$ | $-\frac{1}{3}$ | $-\frac{1}{2}$ |
| 5 | 3.95 | 1.17 | $-\frac{1}{3}$ | $\frac{1}{6}$ | $-\frac{1}{2}$ |
| 6 | $-0.68$ | 0.90 | $-\frac{1}{6}$ | $\frac{1}{2}$ | 0.25 |
| 7 | $-0.10$ | 0.94 | 1 | $\frac{1}{6}$ | $-\frac{1}{6}$ |

Table 4.1: Benchmark points used in Figs. 4.1–4.4. Benchmarks 1, 2, 4 and 6 are updated with respect to the original clusters [25].

shape of the $p_T(h)$ distribution is reproduced by the reweighting procedure (i.e. dips associated with cancellations of triangle- and box-type contributions, and the position of the peaks), discrepancies of up to $\sim 40\%$ can be observed (e.g. in benchmark 1), with smaller deviations of $\mathcal{O}(10-20\%)$ appearing in other benchmark scenarios. The reweighting is performed based on the distribution of the invariant mass $m_{hh}$, which is insensitive to additional jet radiation. Thus the effect of additional radiation is by construction entirely neglected in the reweighted samples. In the exact calculation, on the other hand, the jet emission spectrum will vary significantly depending on which contributions are enhanced in the considered benchmark scenario. For experimental analyses, the closure between the distributions of the final discriminant(s) of a BSM sample and the SM sample reweighted to the same BSM scenario should be studied. Deviations such as the one reported here for $p_T(h)$ should be taken into account through a dedicated uncertainty treatment. Finally, we would like to comment that even if the reweighting would have been performed directly on the $p_T(h)$ distribution we would not have expected a very good agreement between the fully-generated samples and the reweighted ones. The reason is that, in the generation of the extra radiation, Powheg includes a process-dependent Sudakov form factor and hence the dependence on the Wilson coefficients will no longer be polynomial for kinematic variables strongly influenced by the extra radiation.

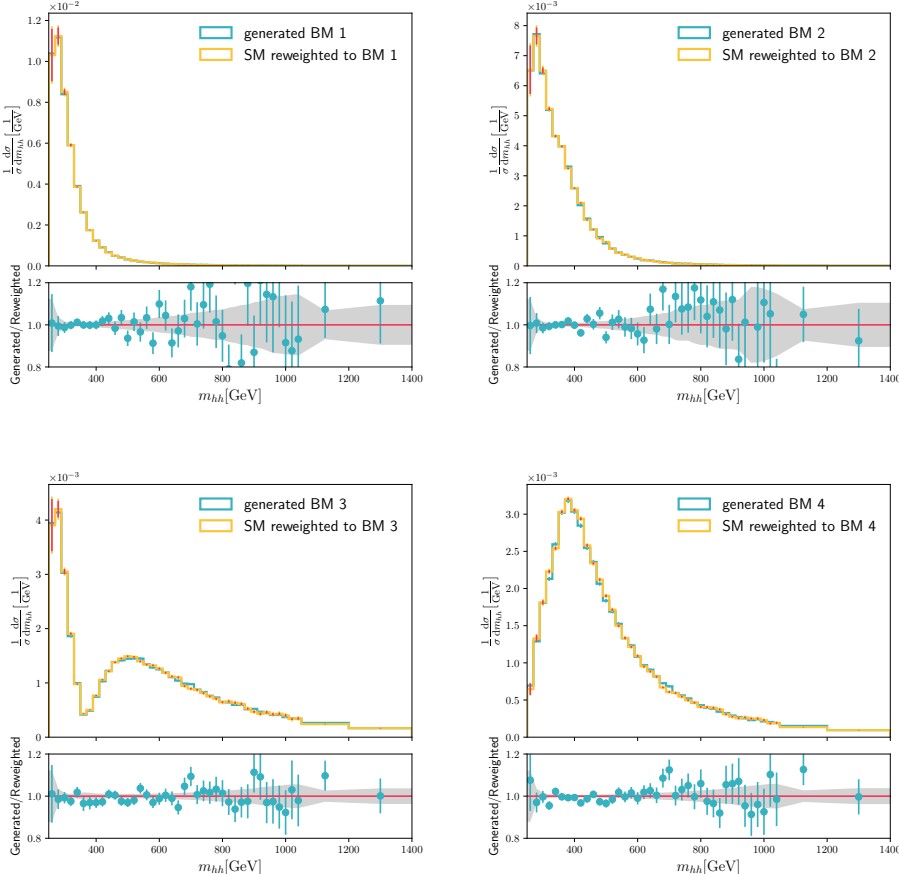

Figure 4.1: Comparison of the $m_{hh}$ distribution of the generated benchmark model (BM) samples 1-4 and the reweighted SM sample. The distributions account for the varying bin width. The bin-by-bin ratio of the generated and reweighted samples is shown in each lower panel. The uncertainties come from the limited number of generated events as well as the reweighting procedure (the latter is shown separately as red error bars in the upper panel and grey bands in the lower panel).

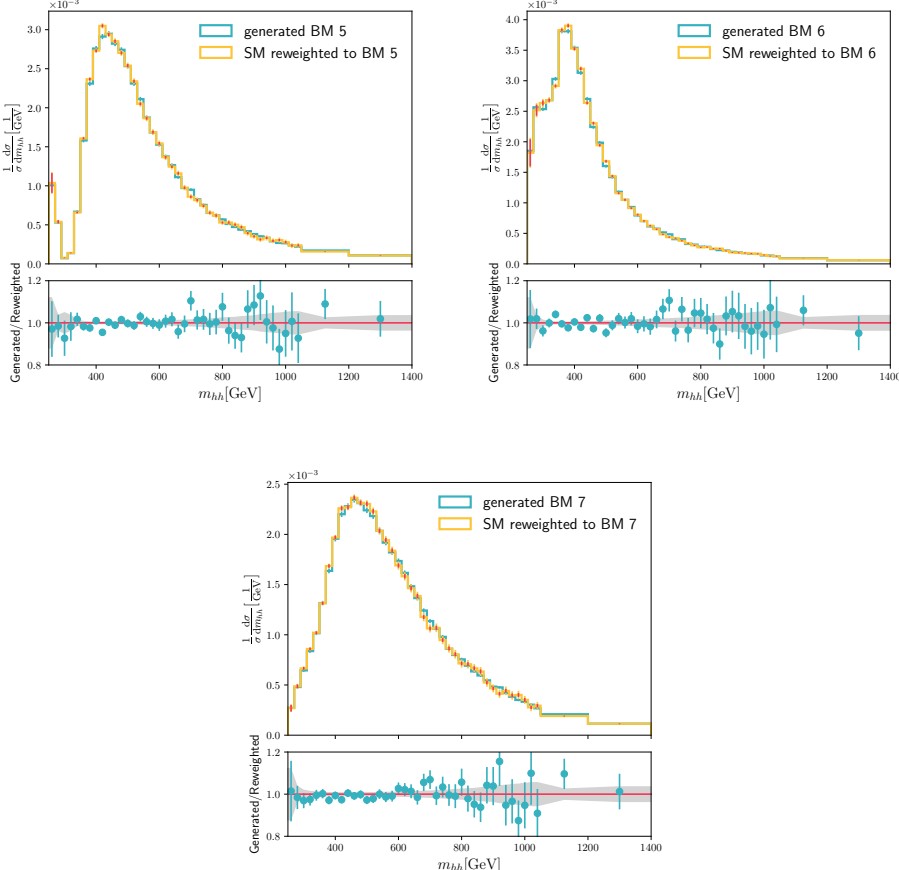

Figure 4.2: Comparison of the $m_{hh}$ distribution of the generated benchmark model (BM) samples 5-7 and the reweighted SM sample. The distributions account for the varying bin width. The bin-by-bin ratio of the generated and reweighted samples is shown in each lower panel. The uncertainties come from the limited number of generated events as well as the reweighting procedure (the latter is shown separately as red error bars in the upper panel and grey bands in the lower panel).

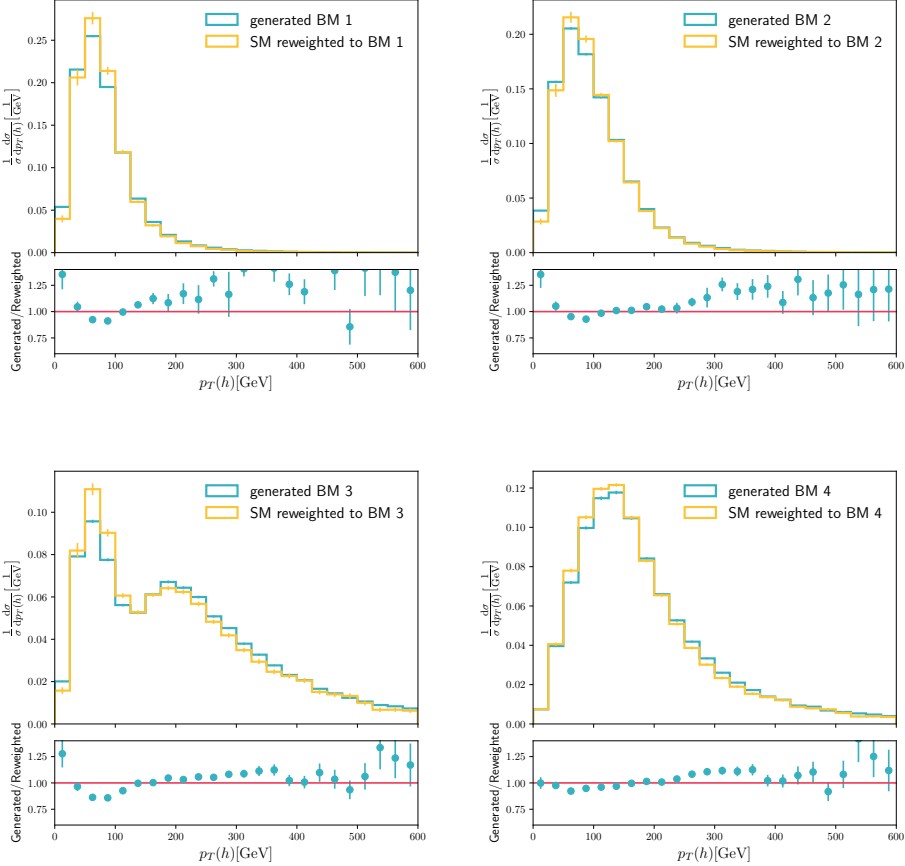

Figure 4.3: Comparison of the $p_T(h)$ distribution of the generated benchmark model (BM) samples 1-4 and the reweighted SM sample. The bin-by-bin ratio of the generated and reweighted samples is shown in each lower panel. Only the uncertainty coming from the limited number of generated events is taken into account.

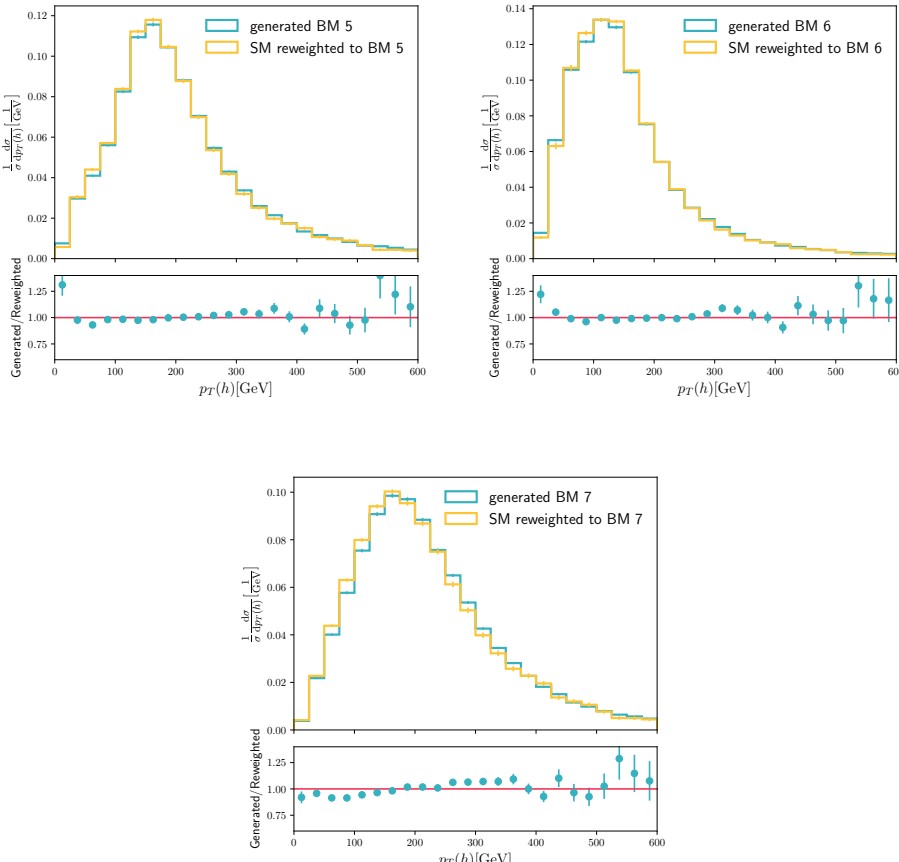

Figure 4.4: Comparison of the $p_T(h)$ distribution of the generated benchmark model (BM) samples 5-7 and the reweighted SM sample. The bin-by-bin ratio of the generated and reweighted samples is shown in each lower panel. Only the uncertainty coming from the limited number of generated events is taken into account.

# Chapter 5

# Conclusions

In this note, we discuss Higgs boson pair production in gluon fusion in HEFT and SMEFT at NLO (with full top quark mass dependence). For the reader's convenience, we recap how the existing NLO tools for this process, i.e. the `POWHEG` implementations `ggHH` [58] and `ggHH_SMEFT` [59, 60], are to be used. In particular, we investigate the potential translation between HEFT and SMEFT at the level of the Lagrangian. We point out that such a translation is extremely delicate as there are various issues to be considered, such as whether or not to include the running $\alpha_s$ into the Wilson coefficients, or how to truncate the EFT expansion.

We also discuss various known sources of uncertainties, namely scale, PDF+$\alpha_s$, top quark renormalisation scheme, EW corrections and statistical uncertainties, the latter being associated to the numerical grid encoding the two-loop virtual corrections as implemented in `POWHEG`, which can be quantitatively different to those present in the SM. For instance, the uncertainty arising from the numerical evaluation of the two-loop virtual corrections can be larger than in the SM case for BSM scenarios which are enhanced in phase-space regions that are not well populated in the SM, if the virtual corrections are substantial in those bins.

In addition, we update the existing kinematic benchmark scenarios to account for recent constraints e.g. on the top quark Yukawa coupling. For these updated scenarios, we show results of a reweighting method that can be used to accelerate experimental analysis significantly. We provide the polynomial coefficients, along with the set of covariance matrices, needed for the reweighting at $\sqrt{s} = 13$ TeV, and discuss comparisons of reweighted samples to dedicated results obtained by running the full event simulation.

## Acknowledgements

We are grateful to all members of the LHC Higgs WG for stimulating discussions that led to this document. We thank in particular E. Bagnaschi, K. Leney, F. Monti and A. Tishelman-Charny for their important comments on the manuscript. T.I.C., L.S.P. and J.S. acknowledge support by the Knut and Alice Wallenberg foundation under the grant KAW 2017.0100 (SHIFT project). The research of G.H. and J.L. was supported by the Deutsche Forschungsgemeinschaft (DFG, German Research Foundation) under grant 396021762 - TRR 257. S.Ö.'s research was funded by the Carl Trygger Foundation through the grant CTS 20:129. L.S. is supported by a Royal Society Research Professorship (RP\R1\180112) and by Somerville College.

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
