# Peer review of "Effective Field Theory descriptions of Higgs boson pair production"

_SciPost Physics Community Reports, doi:SciPost Phys. Comm. Rep. 2 (2024)_

## Round 1 · Referee Report · Anonymous (Referee 1) · 2024-5-10

Strengths

1-Summarises concisely EFT aspects of Higgs pair production
2-It presents the relevant codes.
3-Provides some guidelines for associated theoretical uncertainties

Weaknesses

1-Reweighting procedure suggested is not precise enough.
2-Some clarifications would help at some points in the text.

Report

I have the following comments:

What are the total cross-sections predicted by the various benchmarks points? Aren't some of these already excluded by the experimental bounds on the total HH cross-section?

I am not sure I follow the logic behind modifying the benchmarks by changing Lambda. This is not rescaling the EFT contribution by an overall factors (unless one only looks at the linear contribution only). This will also change its shape. Is this the intention?

On page 12, the authors suggest that option c) contains a subset of operators contributing at dim-8, but option c) only has dimension-6 operators. Are they referring to the possible mixing of dimension-6 double insertions with dimension-8 operators?

The reweighting should be done at a PS point by PS point basis, using the corresponding amplitudes. A reweighting based on just the invariant mass will not be precise enough . This becomes obvious in fig. 4.3. Is a more precise reweighting not an option?

Looking at 4.1, it's clear that one just needs 23 event samples, each one corresponding to a different coupling combination. Then results for any set of the couplings can be extracted by correctly combining those samples. Is it not possible to just produce those 23 samples and thus avoid the need for any reweighting?

Requested changes

In addition to addressing the comments above:

1-I would perhaps rephrase the sentence about SMEFTsim and SmeftFR. As these are tree-level models they cannot be used for HH production at all (even the infinite top mass limit is not available if the effective gghh vertex is not included).

2-I would suggest adding some results for the chromomagnetic operator given that some of the authors have now included this operator to the Powheg implementation. This would complete the SMEFT computation.

Recommendation

Ask for minor revision

  • validity: good
  • significance: good
  • originality: low
  • clarity: high
  • formatting: excellent
  • grammar: excellent

Author:  Ludovic Scyboz  on 2024-07-26  [id 4657]

(in reply to Report 1 on 2024-05-10)
Category:
answer to question

We thank the referee for their comments and provide our replies below.

What are the total cross-sections predicted by the various benchmarks points? Aren't some of these already excluded by the experimental bounds on the total HH cross-section?

We have added an extra column in Table 4.1. listing the corresponding total cross sections. We have also clarified the relation to previous benchmark points that had been derived to take updated constraints into account. At the time of submission of the manuscript to arXiv, the benchmark points were up-to-date with the current constraints. Furthermore, as the experimental shape analysis operates on normalised cross sections, the experimental bounds on the total HH cross section are not considered as a criterion that would strictly forbid these benchmark points. The benchmark points rather serve as a proxy for certain mHH shapes.

I am not sure I follow the logic behind modifying the benchmarks by changing Lambda. This is not rescaling the EFT contribution by an overall factors (unless one only looks at the linear contribution only). This will also change its shape. Is this the intention?

The purpose of changing $\Lambda$ is to show that a translation from a valid HEFT benchmark point can lead out of the validity range for SMEFT values of the Wilson coefficients at low values of $\Lambda$, such as $\Lambda=1$TeV (associated, for example, with negative cross sections in the linearised dimension-6 case). Increasing $\Lambda$ effectively leads to smaller deformations of the SM case, and the shape approaches the SM shape. Therefore, as pointed out by the referee, the shape of a benchmark point changes dramatically as a function of $\Lambda$. It follows that the benchmark points defined in HEFT cannot be used straightforwardly as shape benchmarks in SMEFT.

On page 12, the authors suggest that option c) contains a subset of operators contributing at dim-8, but option c) only has dimension-6 operators. Are they referring to the possible mixing of dimension-6 double insertions with dimension-8 operators?

As the text was seemingly unclear, we have modified it to the following:

"Option (c) includes all terms of option (b) and in addition double insertions of dimension-6 operators. The double insertions formally are of the same order as dimension-8 operators, however they only form a small subset of dimension-8 contributions and also lack ${\cal O}\left(\Lambda^{-4}\right)$ terms following the field redefinition of Eq. (2.3). Therefore, their inclusion can only be useful to get an idea of neglected higher-dimension terms, see also Ref. [83]"

The reweighting should be done at a PS point by PS point basis, using the corresponding amplitudes. A reweighting based on just the invariant mass will not be precise enough. This becomes obvious in fig. 4.3. Is a more precise reweighting not an option?

The current reweighting, which is based only on the invariant mass of the di-Higgs system, does not always perfectly satisfy closure tests as pointed out in particular for the transverse momentum spectrum. Nevertheless, note that the overall shapes are still fairly well described. We argue that what is considered precise enough should be related to the current experimental sensitivity and how the analyses are influenced by other systematic uncertainties. Thus, we suggest that any user should evaluate systematic uncertainties when using this reweighting method. In the manuscript, we explicitly encourage the user to do so, p. 27:

"For experimental analyses, the closure between the distributions of the final discriminant(s) of a BSM sample and the SM sample reweighted to the same BSM scenario should be studied. Deviations such as the one reported here for pT (h) should be taken into account through a dedicated uncertainty treatment.''

It is expected that any non-closure uncertainties originating from the reweighting has a small impact compared to other systematic uncertainties in experimental analyses. Therefore, given the current experimental and theoretical uncertainties, this reweighting is expected to be precise enough for use by the experiments.

One could also consider including the transverse momentum of the di-Higgs system in the reweighting to account for the real radiation. However, as explained in the text, this variable is impacted by the Sudakov form factor from POWHEG, which would lead to a non-polynomial dependence of this variable and the Wilson coefficient. As argued above, the reweighting based on the invariant mass is expected to work well enough given the current experimental uncertainties, but users should evaluate non-closure uncertainties. A refinement of the reweighting, whether based on amplitudes or on an extra distribution, is therefore left for future work.

In the manuscript, we have added the following sentence:

"An alternative would be to add an additional variable to the reweighting, such as the cosine of the angle between one of the Higgs bosons and the beam in the CM frame, which is a suggested option for di-Higgs production via gluon-gluon fusion. However, this variable is expected to be flat for many BSM models. This can be understood from a partial wave analysis: the leading partial wave is independent of $\cos\theta$, and most BSM models are not expected to suppress the leading partial wave substantially for this process. Therefore such a double differential description is not expected to improve the reweighting."

Looking at 4.1, it's clear that one just needs 23 event samples, each one corresponding to a different coupling combination. Then results for any set of the couplings can be extracted by correctly combining those samples. Is it not possible to just produce those 23 samples and thus avoid the need for any reweighting?

The referee is right in pointing this out. The method of using samples is one possibility for extracting the EFT predictions. However, that method is already known and would not provide anything new to the field if we demonstrated it. The drawback of using that method in an experimental analysis is that one would need at least 23 full simulation samples (including full detector simulation), which is computationally expensive to generate. Here, we propose an alternative method, where we extracted our reweighting coefficients form a larger set (63 samples), such that the (MC statistical) uncertainty on the coefficients fitted to the larger set is reduced. Then one can use the reweighting polynomial to reweight one full simulation sample to extract the EFT prediction, thus making our proposed method much less computationally expensive. In the end, it is up to the user to decide which method they prefer.

Requested changes

In addition to addressing the comments above:

1-I would perhaps rephrase the sentence about SMEFTsim and SmeftFR. As these are tree-level models they cannot be used for HH production at all (even the infinite top mass limit is not available if the effective gghh vertex is not included).

We have now pointed out that only the heavy top limit would work with these tools.

2-I would suggest adding some results for the chromomagnetic operator given that some of the authors have now included this operator to the Powheg implementation. This would complete the SMEFT computation.

We have added a comment about the availability of the chromomagnetic operator and four-top operators in the ggHH_SMEFT code, and have added a plot showing the contributions form $C_{tG}$ in Figure 3.4.

---

## Round 1 · Referee Report · Anonymous (Referee 1) · 2024-8-30

Report

The authors have responded to my comments and made appropriate changes in the text where needed.

Recommendation

Publish (meets expectations and criteria for this Journal)

---

## Round 2 · Author Response

Dear Editor,

We would like to resubmit our manuscript "Effective Field Theory descriptions of Higgs boson pair production" including minor revisions as requested by the referee/editor-in-charge. We thank the referee for their comments and have implemented all changes, which are described in our reply to the referee and below, into the resubmitted version.

Best regards,

The authors

---

## Round 2 · List of Changes

- We have added an extra column in Table 4.1. listing the corresponding total cross sections.

- On p. 12, we have modified the paragraph that was unclear, to

"Option (c) includes all terms of option (b) and in addition double insertions of dimension-6 operators. The double insertions formally are of the same order as dimension-8 operators, however they only form a small subset of dimension-8 contributions and also lack O(Λ−4) terms following the field redefinition of Eq. (2.3). Therefore, their inclusion can only be useful to get an idea of neglected higher-dimension terms, see also Ref. [83]"

- On p. 15, we have modified the paragraph about the chromomagnetic and four-tops operators, we have added a comment about their availability in ggHH_SMEFT, and have also added a Figure (3.4, on p. 22) to show the impact of the CtG operator:

"The chromomagnetic operator and four-top operators are subleading from the point of view of weakly interacting UV dynamics. These operators have been included in the ggHH SMEFT code recently [60], with their matrix elements being considered at LO QCD. Combinations with the already presented NLO QCD calculation are possible and follow the power counting rules outlined in Ref. [60]. The effect of the Wilson coefficient CtG on the SM result at LO QCD, using current constraints of Ref. [86], is illustrated in Fig. 3.4. Considering the chromomagnetic operator and the four-top operators in isolation is delicate since the independence on the scheme to continue γ5 to D space-time dimensions can only be shown for a combination of these operators [87]."

- On p. 28, we have added a comment about extensions of the reweighting to include more differential information as a potential alternative to improve the fit description of the Higgs transverse momentum:

"An alternative would be to add an additional variable to the reweighting, such as the cosine of the angle between one of the Higgs bosons and the beam in the CM frame, which is a suggested option for di-Higgs production via gluon-gluon fusion. However, this variable is expected to be flat for many BSM models. This can be understood from a partial wave analysis: the leading partial wave is independent of cosθ, and most BSM models are not expected to suppress the leading partial wave substantially for this process. Therefore such a double differential description is not expected to improve the reweighting."

- We have also updated the references with the correct published version for references which have been published in the meantime (i.e. Refs. [29], [87])

---

## Editorial Decision

published